# Kinetics of High-Sensitive Cardiac Troponin I in Patients with ST-Segment Elevation Myocardial Infarction and Non-ST Segment Elevation Myocardial Infarction

**DOI:** 10.3390/diagnostics15182390

**Published:** 2025-09-19

**Authors:** Adi Haizler, Ranel Loutati, Louay Taha, Mohammad Karmi, Dana Deeb, Mohammed Manassra, Noam Fink, Pierre Sabouret, Jamal S. Rana, Mamas A. Mamas, Ofir Rabi, Akiva Brin, Amro Moatz, Maayan Shrem, Abed Qadan, Nir Levi, Michael Glikson, Elad Asher

**Affiliations:** 1The Jesselson Heart Center, Shaare Zedek Medical Center, Eisenberg R&D Authority, Hebrew University of Jerusalem, Jerusalem 9103102, Israeldanad@szmc.org.il (D.D.);; 2Faculty of Medicine, Hebrew University of Jerusalem, Jerusalem 9112102, Israel; noamfink@bezeqint.net; 3Department of Military Medicine and “Tzameret”, Faculty of Medicine, Hebrew University of Jerusalem, Jerusalem 9112001, Israel; ranellout@gmail.com; 4Assuta Medical Center, Tel Aviv-Yafo 6329302, Israel; 5ACTION Study Group, Institut de Cardiologie, Hôpital Pitié-Salpêtrière, Sorbonne Université, 75005 Paris, France; 6National College of French Cardiologists, 75014 Paris, France; 7Division of Cardiology, Kaiser Permanente Northern California, Oakland, CA 94612, USA; 8Keele Cardiovascular Research Group, Centre for Prognosis Research, Keele University, Newcastle ST5 5BG, UK

**Keywords:** hs cardiac troponin, STEMI, NSTEMI, kinetics

## Abstract

**Background/Objectives:** Existing data regarding the kinetics of cardiac troponin I (cTnI) are limited. The aim of the current study was to evaluate the kinetics of highly sensitive (hs) cTnI following acute myocardial infarction (MI) in a large-scale, real-world cohort. **Methods:** A prospective observational cohort study included all consecutive patients admitted to the intensive cardiovascular care unit (ICCU) with ST-segment elevation MI (STEMI) and non-ST-segment elevation MI (NSTEMI) who underwent percutaneous coronary intervention (PCI) between January 2020 and April 2024. Hs-cTnI concentrations were measured at the time of presentation and daily thereafter. **Results:** A total of 1174 STEMI patients [191 females (16.3%)] with a mean age of 63 years and 767 NSTEMI patients [137 females (17.9%)] with a mean age of 66.7 years were enrolled. The average hs-cTnI peak levels were 77,937.99 ng/L and 24,804.73 ng/L for STEMI and NSTEMI patients, respectively. A single peak of hs-cTnI was observed in 83% and 78% of STEMI and NSTEMI patients, respectively, while two peaks were observed in 11% and 19% and three or more peaks were observed in 6% and 3% of STEMI and NSTEMI patients, respectively. A higher number of peaks was associated with a lower ejection fraction and more in-hospital complications. Additionally, a higher number of peaks correlated with a higher in-hospital mortality rate among NSTEMI patients. **Conclusions:** Most STEMI and NSTEMI patients displayed a monophasic kinetic pattern of hs-cTnI peak levels. However, a greater number of hs-cTnI peaks was linked to a higher incidence of clinical complications, lower ejection fraction, and increased mortality.

## 1. Introduction

Troponin is an essential protein complex primarily found in cardiac muscle, playing a central role in regulating muscle contraction [1]. This complex comprises three subunits: troponin I (cTnI), troponin T (cTnT), and troponin C (cTnC) [2]. Elevated levels of both cTnT and cTnI in the bloodstream typically indicate myocardial injury. Cardiac troponins are the recommended biomarkers for the diagnosis of acute myocardial infarction (MI) according to contemporary guidelines [3]. Nevertheless, while cTnT is predominantly located in cardiac muscle and can be detected in skeletal muscle in smaller quantities, cTnI is highly specific to cardiac muscle [4]. Therefore, cTnI is considered a more specific biomarker for myocardial pathologies compared to cTnT [5]. Other clinical conditions, including heart failure, myocarditis, pulmonary embolism, and cardiac surgery, may also lead to increased troponin levels, reflecting myocardial damage in these scenarios [6].

Studies have highlighted different kinetics of cTnT and cTnI in patients presenting with acute MI [7]. This difference suggests underlying molecular, immunoreactivity, or kidney-clearing pathway differences [8]. Among patients with ST-segment elevation MI (STEMI) undergoing percutaneous coronary intervention (PCI), studies have suggested that cTnT exhibits biphasic kinetics with a notable second peak at around 77 h post-admission, while cTnI usually demonstrates only one peak followed by a log-linear decrease [9]. In non-ST-segment elevation MI (NSTEMI) patients, studies have suggested that cTnI shows higher concentrations compared to cTnT, with cTnI exhibiting fast monophasic elimination and troponin T displaying slow, progressively decreasing elimination [8].

Currently, the literature regarding cTnI kinetics post-MI has predominantly consisted of relatively small-scale studies, with short follow-up periods. These studies have primarily focused on cohorts of either STEMI or non-STEMI cases, with a limited representation of women and the elderly, some of which did not measure high-sensitivity cardiac troponin assays (hs-cTn), thus lacking information regarding women, the elderly, and large-scale contemporary real-world data. Moreover, data concerning the long-term prognosis according to kinetics are limited. Therefore, the aim of the current study was to provide recent data on acute MI patients in a large-scale tertiary care center intensive cardiovascular care unit (ICCU) and to comprehensively characterize cTnI kinetics following acute MI.

## 2. Methods

### 2.1. Study Objectives

The primary objective of this prospective observational study was to characterize the kinetic patterns of hs-cTnI in patients with STEMI and NSTEMI undergoing PCI. Secondary objectives included (1) quantifying the number and timing of troponin peaks; (2) comparing kinetic profiles between STEMI and NSTEMI groups; and (3) assessing clinical outcomes, specifically left ventricular ejection fraction (LVEF), from in-hospital complications (e.g., no reflow, stent thrombosis, and arrhythmias), as well as in-hospital mortality. All outcomes were prospectively collected from electronic medical records. LVEF was assessed using echocardiography within 48 h of admission. In-hospital complications and mortality were recorded and confirmed through attending physician documentation and hospital discharge summaries.

### 2.2. Study Population

A prospective single-center observational cohort study was performed in the ICCU of Shaare Zedek Medical Center: a tertiary care center that treats more than 1000 patients with cardiovascular disease every year.

Inclusion criteria: All consecutive patients diagnosed with STEMI and NSTEMI who were admitted to the ICCU and underwent PCI between January 2020 and April 2024 were included.

Exclusion Criteria: Patients with STEMI and NSTEMI who did not undergo PCI for any reason were excluded. The diagnosis of STEMI was based on chest pain lasting for more than 30 min associated with an ST-segment elevation of more than 2 mm on ECG in two adjacent derivations [3].

The diagnosis of NSTEMI was based on symptoms of myocardial ischemia, new ECG ischemic changes, and a rising and/or falling pattern of hs-cTn with at least one value above the 99th percentile URL according to the ESC guidelines for acute coronary syndrome (ACS) [3].

The study was conducted in accordance with the Declaration of Helsinki and approved by the Institutional Review Board of the Shaare Zedek Medical Center (protocol code 0260-24-SZMC). Informed consent was waived by the IRB due to the observational design of the study.

### 2.3. Data Collection

Data was anonymously documented in the ICCU by the local coordinator and prospectively submitted into an electronic case report form. Data was checked for accuracy and out-of-range values using the coordinating unit. Demographic data, presenting symptoms, comorbid conditions, and findings from a physical examination were systematically recorded. Laboratory, imaging, angiographic results, and clinical course data were collected as well.

### 2.4. Measurement of Hs-cTnI

Hs-cTnI testing was conducted upon arrival at the hospital and was repeated daily, at least 3 times during admission, in accordance with national guidelines [10,11]. In accordance with the cardiac intensive care unit protocol at Shaare Zedek Medical Center, hs-cTnI levels were measured every 12 h in clinically stable patients and every 8 h in critically ill patients. Additional post-procedural measurements were sometimes performed at the discretion of the treating cardiologist to evaluate possible periprocedural myocardial injury. Patients were then followed up 1 year after presentation. All reported variables were taken from the index hospitalization. Hs-cTnI assays were determined in a central laboratory (using ARCHITECT STAT hs-cTnI immunoassay) with a 99th percentile reference level of 17 ng/L for females and 35 ng/L for males [12]). All patients who were admitted to the ICCU and had at least three hs-cTnI tests, and the highest hs-cTnI measurement was selected as the peak hs-cTnI level for the purpose of this study, regardless of measurement timing. According to the criteria outlined in the Fourth Universal Definition of Myocardial Infarction [2], a second troponin peak was defined as a renewed and significant rise in troponin levels following an initial decline or plateau, exceeding the expected analytical variation. These criteria included (1) an initial sharp rise in troponin concentrations, (2) followed by a partial decline or stabilization, and (3) a subsequent distinct increase. Consistent with previous studies [13] investigating the kinetics of troponin following acute MI, a second peak of hs-cTnI was defined as an increase of at least 15% in troponin concentration after an initial decline from the first peak value.

### 2.5. Coronary Angiography and PCI

Coronary angiography and PCI were performed to determine coronary perfusion according to the TIMI criteria. Primary PCI was performed with standard methods if the coronary anatomy was deemed suitable for angioplasty at the discretion of the interventional cardiologist following guideline recommendations [14,15].

### 2.6. Statistical Analysis

Continuous variables were expressed as the mean ± standard deviation if normally distributed or the median with interquartile range if skewed. Categorical variables were presented as frequencies and percentages. The association between two categorical variables was tested using either the Chi-square test or Fisher’s exact test. Comparing quantitative variables between two independent groups was performed by applying either the two-sample *t*-test or the non-parametric Mann–Whitney test. The strength of the association between two quantitative variables was assessed by calculating the Spearman non-parametric correlation coefficient. The non-parametric tests were applied to data that was not normally distributed. The multivariate linear regression model (ANCOVA) was applied to simultaneously assess the effect of several independent variables on a quantitative dependent variable. The model was applied using stepwise, forward, and likelihood ratio methods. Logistic regression analysis was performed separately for STEMI and NSTEMI patient groups to identify independent predictors of multiple troponin peaks (defined as ≥2). Variables included in the models were selected based on clinical relevance and results from univariate analyses. Model performance was assessed using receiver operating characteristic (ROC) curve analysis, with the area under the curve (AUC) reported as a measure of discrimination. All statistical tests applied were two-tailed, and a *p*-value of 0.05 or less was considered statistically significant. Analysis was performed using the statistical program IBM SPSS Statistics version 30 (https://www.ibm.com/support/pages/downloading-ibm-spss-statistics-30, accessed on 13 June 2025) and R software version 3.4.4 (https://cran-archive.r-project.org/bin/windows/base/old/3.4.4/, accessed on 13 June 2025) (R Foundation for Statistical Computing).

## 3. Results

### 3.1. Baseline Characteristics

A total of 2527 patients were included during the study period. Of them, 550 (21.7%) did not undergo PCI and, hence, were excluded. Another 36 (1.4%) patients were excluded due to a lack of data. The remaining 1941 (76.8%) patients comprised the final cohort, including 1174 (60.5%) STEMI and 767 (39.5%) NSTEMI patients (Figure 1).

### 3.2. Patient Characteristics

NSTEMI patients were older than STEMI patients (67 years old vs. 63 years old, respectively, *p* < 0.001), while sex distribution (82.1% vs. 83.7% male, *p* = 0.353) and body mass index (mean 28) were similar. NSTEMI patients had a higher prevalence of cardiovascular risk factors, including hypertension, dyslipidemia, diabetes mellitus (DM), a history of coronary artery disease (CAD), coronary artery bypass graft surgery (CABG), peripheral artery disease (PAD), and congestive heart failure (CHF), as shown in Table 1. In contrast, smoking (44.5% vs. 34.7%, *p* < 0.001) and a family history of CAD (12.5% vs. 9.4%, *p* = 0.033) were significantly more prevalent in STEMI patients. Left ventricular ejection fraction (LVEF) was lower in STEMI patients compared to NSTEMI patients (45.8% vs. 48.9%, respectively, *p* < 0.001). There were no significant differences between groups in the prevalence of cerebro-vascular accident (CVA)/transient ischemic attack (TIA), chronic obstructive pulmonary disease (COPD)/asthma, or pulmonary embolism (PE). All patients who underwent PCI were treated according to standard clinical protocols, including the administration of antiplatelet therapy and statins. Additionally, most patients received beta-blockers and ACE inhibitors as part of their medical management.

### 3.3. Complications and Outcomes

During the course of hospitalization, STEMI patients had a higher complication rate of CHF (5.3% vs. 3.1%, *p* = 0.024), LV thrombus (2.4% vs. 0.7%, *p* = 0.004), and in-hospital mortality (3.0% vs. 1.2%, *p* = 0.009). In contrast, NSTEMI patients required more blood transfusions (2.7% vs. 1.3%, *p* = 0.02). The incidence of cardiogenic shock was 4.7% vs. 3.8%, *p* = 0.339. Mechanical complications, including ventricular septal rupture (VSR) or myocardial rupture (0.8% vs. 0.3%, *p* = 0.218), stroke/TIA (1.9% vs. 1.3%, *p* = 0.335), reinfarction (0.4% vs. 0.7%, *p* = 0.529), and significant bleeding (2.9% vs. 3.0%, *p* = 0.896) did not significantly differ between groups. Clinical outcomes and in-hospital complications are summarized in Table 2.

### 3.4. Kinetics of Hs-cTnI

In the STEMI group, 976 (83%) patients exhibited a single hs-cTnI peak, while 129 patients (11%) had two peaks, and 69 (6%) patients had three or more peaks. Similarly, in the NSTEMI group, 600 (78%) patients exhibited a single peak, while 142 (19%) patients had two peaks, and 25 (3%) patients had three or more peaks (Figure 2 and Figure 3).

Figure 2 and Figure 3. Kinetics of hs-cTnI concentrations during the first 100 h after the initial measurement. Thin lines represent individual patients, and the bold line shows the group-level loess-smoothed trend with a 95% confidence interval. The y-axis is truncated at the 99th percentile to minimize the influence of extreme outliers.

### 3.5. Time to Peak and Level of the Hs-cTnI Peak

STEMI patients exhibited higher mean peak hs-cTnI levels than NSTEMI patients (77,937.99 ng/L ± 173,762.86 vs. 24,804.73 ng/L ± 56,985.37, respectively, *p* < 0.001). The time to peak for hs-cTnI was also shorter in STEMI patients (12 h vs. 13 h, *p* = 0.03).

Lastly, STEMI patients underwent more hs-cTnI measurements on average than NSTEMI patients (6.10 ± 5.28 vs. 5.21 ± 4.77, respectively, *p* < 0.001).

### 3.6. Association Between Number of Hs-cTnI Peaks and Patient Baseline Characteristics

Univariate analysis showed no significant association between the number of hs-cTnI peaks and baseline characteristics, including BMI, CVA or TIA, COPD, DM, DLP, family history of CAD, prior CAD, prior CABG, PE, pulmonary hypertension, and smoking. Nevertheless, HTN, PAD, and CHF were associated with an increased number of peaks in NSTEMI patients.

Low LVEF was correlated with a higher number of hs-cTnI peaks (r = −0.144, *p* < 0.001). Moreover, cardiogenic shock was strongly associated with a greater number of hs-cTnI peaks in both STEMI and NSTEMI patients (2.05 vs. 1.24, *p* < 0.001 and 3.13 vs. 1.24, respectively, *p* < 0.001).

Multiple linear regression analyses were performed to identify predictors of the number of hs-cTnI peaks. The models demonstrated adjusted R^2^ values of 0.197 and 0.265 for STEMI and NSTEMI, respectively. In both STEMI and NSTEMI groups, the delayed troponin peak (OR = 1.019, 95% CI 1.017–1.021, B = 0.019, *p* < 0.001; OR = 1.008, 95% CI 1.006–1.010, B = 0.008, *p* < 0.001, respectively) and presence of cardiogenic shock (OR = 1.850, 95% CI 1.460–2.344, B = 0.615, *p* < 0.001; OR = 7.144, 95% CI 5.326–9.577, B = 1.966, *p* < 0.001, respectively) were consistently associated with a higher number of hs-cTnI peaks. Among NSTEMI patients, a history of ischemic heart disease (OR = 1.474, 95% CI 1.149–1.891, B = 0.388, *p* = 0.002) and the presence of CHF, CMP, or MVR (OR = 1.239, 95% CI 1.024–1.499, B = 0.214, *p* = 0.028) were also significant predictors of the number of troponin peaks. Among STEMI patients, reduced EF (OR = 0.985, 95% CI 0.977–0.993, B = −0.015, *p* < 0.001) and higher peak troponin levels (OR = 1.000, 95% CI 1.000–1.000, B < 0.001, *p* = 0.013) were also identified as predictors of an increased number of troponin peaks. The results are summarized in Table 3. Logistic regression models were constructed separately for STEMI and NSTEMI patients to identify predictors of multiple troponin peaks (defined as ≥2). For STEMI patients, the model included hours to the peak troponin level, cardiogenic shock, EF, and peak troponin level. In NSTEMI patients, predictors included cardiogenic shock, hours to the peak troponin level, ischemic heart disease, presence of CHF, CMP, or MVR, and peak troponin level. Model performance was assessed by receiver operating characteristic (ROC) curve analysis, demonstrating good discrimination with an area under the curve (AUC) of 0.74 for STEMI and 0.73 for NSTEMI. The ROC curves are illustrated in Figure 4 and Figure 5.

### 3.7. Association Between Number of Hs-cTnI Peaks and Clinical Complications

Clinical complications were significantly associated with an increased number of hs-cTnI peaks for both STEMI and NSTEMI patients: (2.55 vs. 1.25 and 2.10 vs. 1.27, *p* = 0.001) for cardiogenic shock in STEMI and NSTEMI patients, respectively; (2.76 vs. 1.23 and 2.33 vs. 1.27, *p* = 0.001) for CHF in STEMI and NSTEMI patients, respectively; (2.91 vs. 1.28 and 3.40 vs. 1.27, *p* = 0.003) for post-procedural stroke or TIA in STEMI and NSTEMI patients, respectively; (3.33 vs. 1.30, *p* = 0.009 and 2.36 vs. 1.29, *p* = 0.032) for stent thrombosis in STEMI and NSTEMI patients, respectively; (2.62 vs. 1.27, *p* < 0.001 and 2.13 vs. 1.28, *p* = 0.048) for significant bleeding in STEMI and NSTEMI patients, respectively; and (2.80 vs. 1.30, *p* =0.003; and 3.4 vs. 1.29, *p* < 0.001) for reinfarction in STEMI and NSTEMI patients, respectively. In-hospital mortality was significantly associated with an increased number of peaks in NSTEMI patients but not in STEMI patients (3 vs. 1.28, *p* = 0.033 and 1.43 vs. 1.31, *p* = 0.274, respectively). LV thrombus and blood transfusion were significantly associated with an increased number of peaks in STEMI patients, but were not statistically significant in NSTEMI patients, while mechanical complications (free wall rupture) were not significantly associated with an increased number of peaks in either STEMI or NSTEMI patients. Table 4 summarizes the association between the number of hs-cTnI peaks and clinical complications.

## 4. Discussion

Our contemporary study reports several findings in ACS: (1) The majority of STEMI and NSTEMI patients exhibited a monophasic hs-cTnI kinetic pattern, with a single peak observed in 83% and 78% of patients, respectively. (2) A second and/or third peak was observed in a minority of patients and was associated with an increased risk of adverse events and a higher mortality rate. (3) LVEF exhibited a significant inverse correlation with the number of peaks, suggesting that a greater number of peaks may reflect larger infarcts and greater myocardial injury. (4) The presence of clinical complications, including cardiogenic shock, heart failure, stroke, stent thrombosis, major bleeding, and reinfarction, was significantly associated with an increased number of peaks in both STEMI and NSTEMI patients.

Our large-scale study supports prior pilot studies demonstrating that most patients with STEMI and NSTEMI exhibit a single troponin peak, reinforcing the monophasic kinetic pattern of troponin release following acute MI. Consistent with our findings, Solecki et al. (2015) reported that hs-cTnI levels peak at around 10.9 h post-PCI and then decline in a log-linear fashion, further validating the predominance of a single-peak pattern [7]. Additionally, Van-Doorn et al. (2020) [8] investigated hs-cTnI and hs-cTnT kinetics in NSTEMI patients and identified distinct clearance patterns between these biomarkers. Their study showed that hs-cTnT follows a biphasic release pattern with a delayed secondary peak, whereas hs-cTnI typically follows a monophasic trajectory, which may explain why our study, which focused on hs-cTnI, observed a predominance of single peaks with only a subset of multiple peaks [8].

Importantly, our study suggests that multiple peaks are strongly associated with poorer LVEF, greater myocardial damage, and increased adverse outcomes. Specifically, we identified a subset of patients with multiple troponin peaks, which were significantly linked to lower LVEF, higher complication rates, and increased mortality. This finding aligns with Hartikainen et al. (2022), who observed a biphasic hs-cTnT pattern in a subset of MI patients and found that a second troponin peak was associated with larger infarcts [13]. However, unlike our study, they did not identify a significant association between a second hs-cTnT peak and worse clinical outcomes; this could be explained by the smaller sample size used in their study and the fact that it was conducted in the emergency department and not in the ICCU. This discrepancy may be attributable to differences between hs-cTnI and hs-cTnT kinetics, as hs-cTnT typically exhibits delayed secondary release, while hs-cTnI follows a more rapid clearance pattern. Moreover, our study reinforces the recent study by Loutati et al. (2024) on the relationship between peak troponin levels and mortality and adds new findings to the prognostic value of the number of troponin peaks and mortality rate [16]. These new findings can help to better identify a subgroup of patients at very high risk who may require more intensive management. Further research is warranted to confirm our findings. Although our study did not specifically examine periprocedural myocardial injury (PMI), the study by Armillotta et al. (2024) found that a secondary rise in troponin levels was associated with adverse clinical outcomes in NSTEMI patients [17]. These findings support the potential prognostic importance of biphasic troponin kinetics, warranting further investigation.

Our study offers several distinctive contributions. First, it is a prospective study with a large cohort of both STEMI (1174) and NSTEMI (767) patients, allowing for a detailed evaluation of troponin kinetics across different MI subtypes. Second, we included only ICCU-admitted patients, ensuring the inclusion of a high-risk and clinically uniform population, whereas previous studies often include patients with varying severities. Third, we focused solely on patients who underwent PCI, excluding non-PCI ACS cases due to their distinct troponin kinetics. In reperfused patients, opening the infarct-related artery leads to a rapid “wash-out” of intracellular troponin, while in non-reperfused patients, release and clearance occur more gradually. This approach allowed us to maintain a homogeneous cohort and reduce potential confounding [18]. Finally, the use of hs-cTnI assays enabled a more precise characterization of troponin kinetics, in contrast to prior studies that rely on less sensitive (non-hs) conventional troponin assays. Notably, our study included a larger representation of women (16.3% STEMI, 17.9% NSTEMI) compared to previous research, addressing a persistent gap in cardiovascular studies.

### 4.1. Clinical Implications

The association between multiple hs-cTnI peaks and worse clinical outcomes underscores the potential role of troponin kinetics in refining risk stratification in acute MI. Our findings demonstrate that patients with two or more troponin peaks had significantly higher rates of cardiogenic shock, post-procedural heart failure, stroke or TIA, stent thrombosis, significant bleeding, and reinfarction in both STEMI and NSTEMI populations. Importantly, in NSTEMI patients, multiple peaks were also associated with an increased rate of in-hospital mortality, highlighting their potential as an early indicator of poor prognosis. These results suggest that beyond a single peak value, the pattern and number of troponin peaks could serve as a dynamic marker of ongoing myocardial injury or procedural complications. The observation of biphasic troponin release patterns, particularly a second peak after PCI, was associated with worse clinical features. These associations suggest that troponin kinetics may reflect underlying pathophysiological processes, including reperfusion injury, procedural complications, or evolving myocardial damage. Recognizing such patterns could help identify higher-risk patients who may benefit from intensified monitoring, early imaging (e.g., echocardiography or MRI), or tailored post-discharge care. While these hypotheses warrant prospective validation, they can contribute by refining risk stratification strategies beyond static troponin thresholds. Clinically, this could prompt a shift toward the routine evaluation of hs-cTnI kinetics over time, with greater emphasis on serial measurements post-PCI. Troponin kinetics could also be integrated into prognostic models to guide discharge planning and post-discharge follow-up intensity. Importantly, our findings suggest that the occurrence of a second hs-cTnI peak functions primarily as a marker of underlying pathophysiological processes such as reperfusion injury, procedural complications, or evolving myocardial damage rather than a direct mechanism of myocardial injury. However, as this study is primarily descriptive and exploratory, these associations should be interpreted with caution, and further prospective research is needed to confirm clinical significance. Broader implementation would require multicenter validation and prospective studies to determine whether interventions based on troponin kinetic profiles translate into improved clinical outcomes.

### 4.2. Study Limitations

Our study has several limitations: (1) Troponin measurements were not obtained at strictly uniform intervals, raising the possibility that some peak measurements may have been missed between two consecutive tests, which could have influenced the characterization of troponin kinetics. (2) This was a single-center study, which may limit the generalizability of our findings to broader populations. (3) The follow-up period was relatively short, restricting the ability to assess long-term clinical outcomes. (4) We exclusively measured hs-cTnI, without evaluating troponin cTnT, which may provide additional insights into troponin kinetics. (5) The exact time to catheterization was not recorded for each patient, which may have influenced the observed troponin kinetics. However, in our institution, the majority of STEMI patients underwent primary PCI with a door-to-balloon time of less than 90 min, and most NSTEMI patients were catheterized within 24 h, which aligns with current ESC guidelines [3]. The precise time from the onset of acute chest pain to hospital arrival was not consistently documented. Symptom onset was based on patient self-report, which is inherently subject to recall bias, limiting our ability to accurately establish the clinical onset of myocardial infarction. (6) Regarding the definition of a second troponin peak, we used a threshold of a ≥15% relative increase based on previously published data. However, other expert consensus suggests a ≥20% change as the threshold for clinical significance, especially when initial troponin values exceed the 99th percentile upper reference limit (URL) [2]. This variation in definition could influence the classification of biphasic troponin kinetics. To address this, we performed sensitivity analyses applying the 20% threshold, which demonstrated consistent findings and further support the validity of our results. (7) Although our study provides novel insights into the prognostic value of multiple hs-cTnI peaks, the findings are primarily descriptive and exploratory; hence, further studies are warranted to confirm their causal significance and clinical applicability. (8) Our cohort reflects admissions to the CCU, where STEMI patients are preferentially admitted for urgent reperfusion. In contrast, many lower-risk NSTEMI patients, frequently managed on general cardiology wards due to limited CCU bed capacity, were underrepresented in our dataset. This admission pattern likely contributed to the higher proportion of STEMI and may limit the generalizability of our findings.

## 5. Conclusions

This study provides a comprehensive analysis of hs-cTnI kinetics in a large cohort of STEMI and NSTEMI patients undergoing PCI. While the majority of patients exhibited a monophasic troponin pattern, a subset of patients demonstrated multiple peaks, which were significantly associated with lower LVEF, larger infarcts, increased complication rates in both STEMI and NSTEMI patients, and a higher mortality rate in NSTEMI patients. These findings underscore the prognostic value of hs-cTnI kinetics and suggest that dynamic troponin patterns may provide additional insights for risk stratification and post-MI management. Future studies are needed to explore the clinical utility of incorporating troponin kinetics into prognostic models to optimize patient outcomes.

## Figures and Tables

**Figure 1 diagnostics-15-02390-f001:**
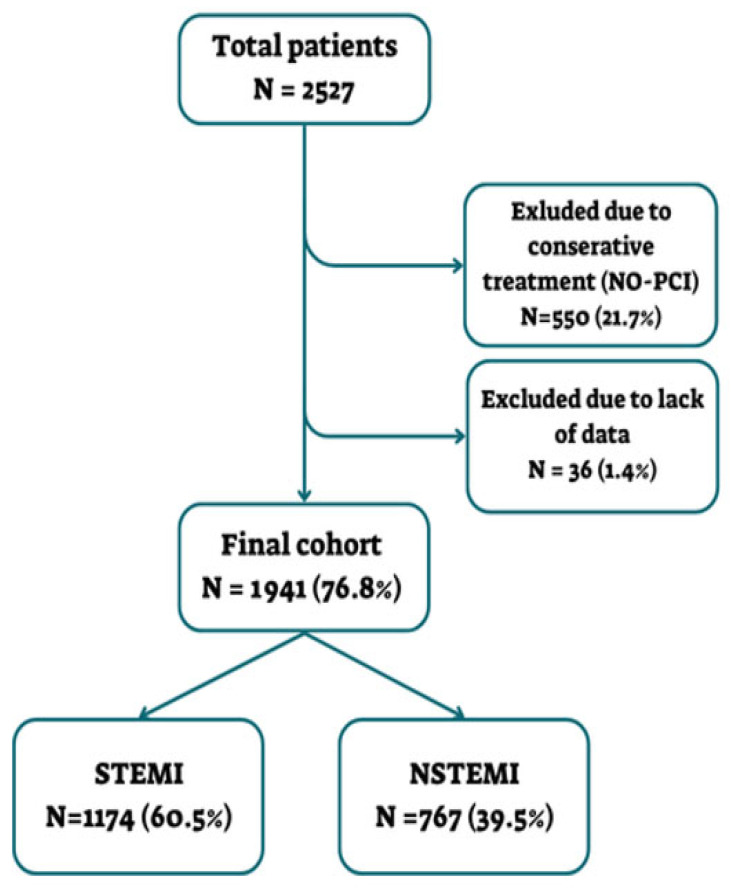
Patient enrollment scheme.

**Figure 2 diagnostics-15-02390-f002:**
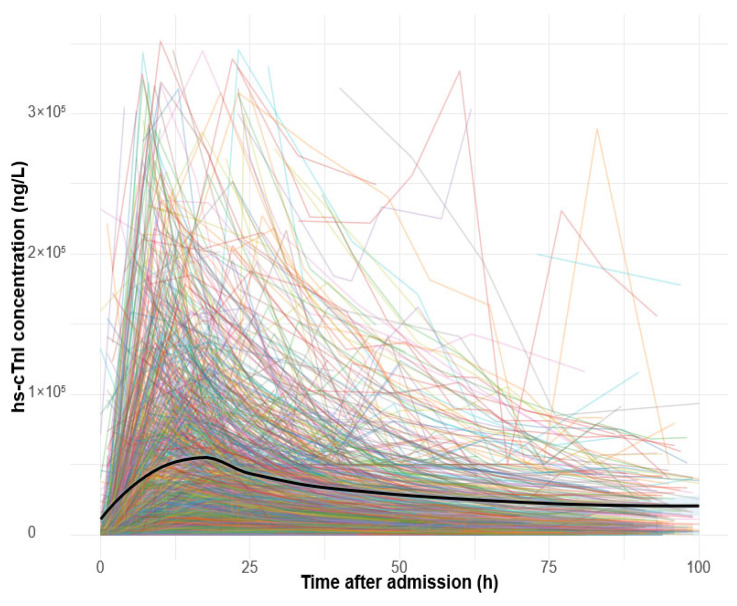
Hs-cTnl kinetics for STEMI patients.

**Figure 3 diagnostics-15-02390-f003:**
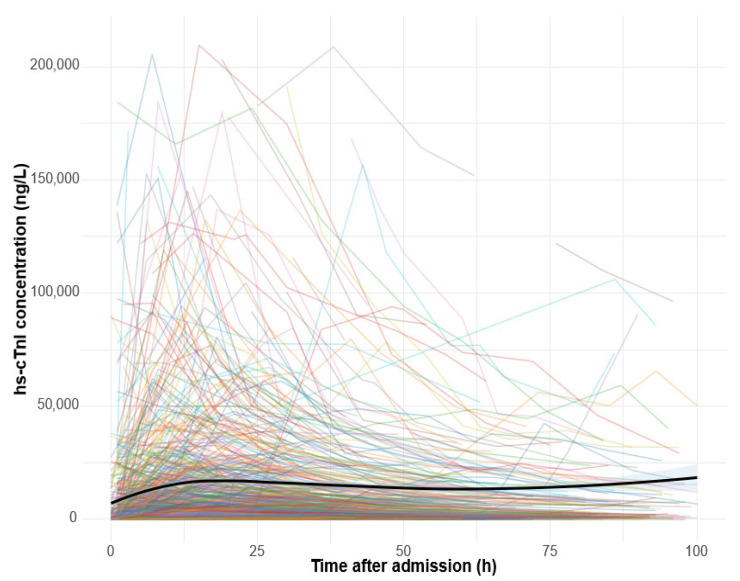
Hs-cTnl kinetics for NSTEMI patients.

**Figure 4 diagnostics-15-02390-f004:**
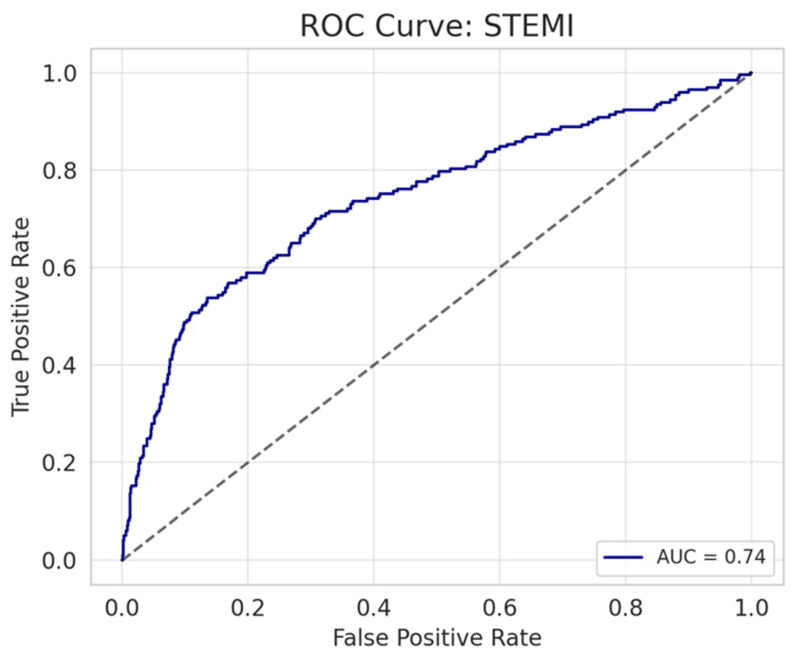
ROC Curve in STEMI patients.

**Figure 5 diagnostics-15-02390-f005:**
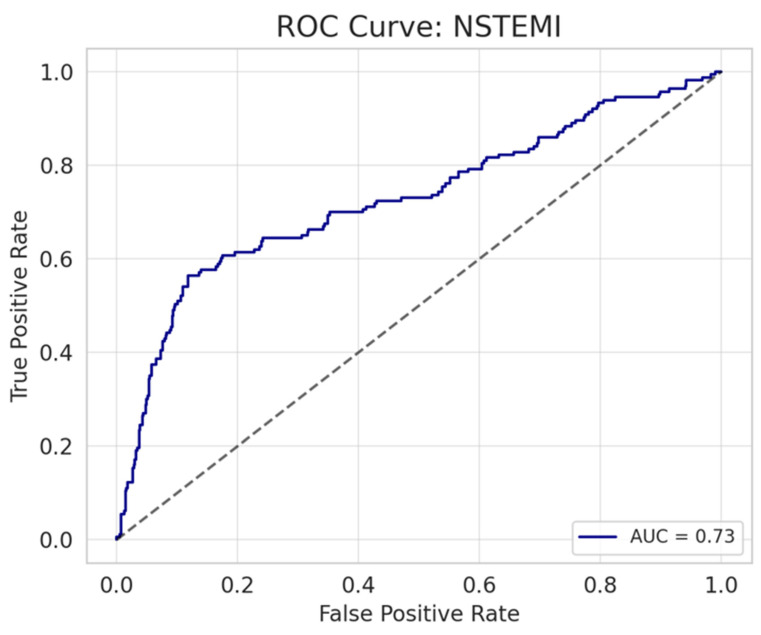
ROC Curve for NSTEMI patients.

**Table 1 diagnostics-15-02390-t001:** Patient characteristics.

Characteristics	Overall (N = 1941)	NSTEMI (N = 767)	STEMI (N = 1174)	*p*-Value
Age, mean (IQR)	65 (57–77)	67 (58–79)	63 (56–75)	<0.001
BMI, mean (kg/m^2^)	28	28	28	0.876
Male gender, %	83.1%	82.1%	83.7%	0.353
EF, median (IQR)	47.22% (41–52)	48.89% (44–54)	45.81% (39–49)	<0.001
Hospital length of stay (LOS), median (IQR)	2.0 days (1–3)	2 days (1–3)	3 days (1–4)	0.02
Hypertension	57.2%	66.1%	51.4%	<0.001
Dyslipidemia	57.1%	62.5%	53.4%	<0.001
Diabetes mellitus	37.4%	42.0%	34.3%	0.001
CAD	30.2%	40.9%	23.3%	<0.001
Prior CABG	5.3%	9.5%	2.5%	<0.001
PAD	4.1%	7.2%	2.1%	<0.001
Heart failure	5.3%	7.4%	3.8%	0.001
Pulmonary hypertension	0.9%	1.7%	0.3%	0.002
Smoking	40.9%	34.7%	44.5%	<0.001
Family history of CAD	11.3%	9.4%	12.5%	0.033
CVA/TIA	5.2%	6.0%	4.6%	0.173
COPD/asthma	5.5%	6.0%	5.2%	0.449
Pulmonary embolism	0.6%	0.9%	0.4%	0.238

BMI = Body Mass Index; EF = Ejection Fraction; CABG = Coronary Artery Bypass Graft; CAD = Coronary Artery Disease; PAD = Peripheral Arterial Disease; CVA = Cerebro-Vascular Accident; TIA= Transient Ischemic Attack; COPD = Chronic Obstructive Pulmonary Disease.

**Table 2 diagnostics-15-02390-t002:** Clinical outcomes and in-hospital complications.

	Overall (N = 1941)	NSTEMI (N = 767)	STEMI (N = 1174)	*p*-Value
Cardiogenic Shock	4.3%	3.8%	4.7%	0.339
Heart Failure	4.4%	3.1%	5.3%	0.024
Mechanical Complications (VSR/Rupture)	0.6%	0.3%	0.8%	0.218
LV Thrombus	1.7%	0.7%	2.4%	0.004
Stroke/TIA	1.6%	1.3%	1.9%	0.335
Reinfarction	0.5%	0.7%	0.4%	0.529
Stent Thrombosis	0.9%	1.4%	0.5%	0.033
Major Bleeding	2.9%	3.0%	2.9%	0.896
In-Hospital Mortality	2.3%	1.2%	3.0%	0.009

VSR = ventricular septal rupture; LV thrombus = left ventricular thrombus; TIA = transient ischemic attack.

**Table 3 diagnostics-15-02390-t003:** Multiple linear regression analyses to identify predictors of the number of hs-cTnI peaks.

STEMI						NSTEMI					
Predictor	β	SE	OR	95% CI	*p* value	Predictor	β	SE	OR	95% CI	*p* value
Hours to Peak Troponin Level	0.019	0.001	1.019	[1.017, 1.021]	<0.001	Hours to Peak Troponin Level	0.008	0.001	1.008	[1.006, 1.010]	<0.001
Cardiogenic Shock	0.615	0.122	1.85	[1.460, 2.344]	<0.001	Cardiogenic Shock	1.966	0.153	7.144	[5.326, 9.577]	<0.001
Peak Troponin Level	<0.001	<0.001	~1.000	[1.000, 1.000]	0.013	Peak Troponin Level	<0.001	<0.001	~1.000	[0.999, 1.000]	0.028
EF	−0.015	0.004	0.985	[0.977, 0.993]	<0.001	IHD	0.388	0.125	1.474	[1.149, 1.891]	0.002
						CHF/CMP/MVR	0.214	0.097	1.239	[1.024, 1.499]	0.028
Constant		0.171			<0.001	Constant		0.033			<0.001

CHF = Congestive Heart Failure; CMP = Cardiomyopathy; EF = Ejection Fraction; IHD = Ischemic Heart Disease; MVR = Mitral Valve Replacement.

**Table 4 diagnostics-15-02390-t004:** Association between number of hs-cTnI peaks and clinical complications.

Clinical Complication	Number of hs-cTnI Peaks in NSTEMI (Complication vs. No Complication)	*p*-Value NSTEMI	Number of hs-cTnI Peaks in STEMI (Complication vs. No Complication)	*p*-Value STEMI
Cardiogenic Shock	2.10 vs. 1.27	0.001	2.55 vs. 1.25	<0.001
Post-Procedural HF	2.33 vs. 1.27	0.001	2.76 vs. 1.23	<0.001
Post-Procedural Stroke/TIA	3.40 vs. 1.27	0.003	2.91 vs. 1.28	<0.001
Stent Thrombosis	2.36 vs. 1.29	0.032	3.33 vs. 1.30	0.009
Significant Bleeding	2.13 vs. 1.28	0.048	2.62 vs. 1.27	<0.001
Re-Infarction	3.40 vs. 1.29	<0.001	2.80 vs. 1.30	0.003
LV Thrombus	1.20 vs. 1.30	0.896	2.61 vs. 1.28	0.001
Blood Transfusion	1.71 vs. 1.29	0.128	2.33 vs. 1.30	<0.001
VSR/Septal Rupture	1.00 vs. 1.30	0.457	1.00 vs. 1.31	0.177
In-Hospital Mortality	3 vs. 1.28	0.033	1.43 vs. 1.31	0.274

HF = Heart Failure; TIA= Transient Ischemic Attack; LV thrombus = Left Ventricular thrombus; VSR= Ventricular Septal Rupture.

## Data Availability

The data presented in this study are available on request from the corresponding author due to institutional and ethical guidelines.

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
