# Peer review of "Kinetics of High-Sensitive Cardiac Troponin I in Patients with ST-Segment Elevation Myocardial Infarction and Non-ST Segment Elevation Myocardial Infarction"

_diagnostics, 2025, doi:10.3390/diagnostics15182390_

Round 1
Reviewer 1 Report
Comments and Suggestions for Authors
This manuscript presents a well-structured and timely prospective observational cohort study investigating hs-cTnI kinetics in a large population of STEMI and NSTEMI patients undergoing PCI. The study addresses an important gap in the literature by characterizing troponin I release patterns in a real-world high-risk population. The results are statistically robust and clinically relevant, offering novel insights into the prognostic value of multiple hs-cTnI peaks. However, there are some areas where the manuscript can be improved for clarity, scientific rigor, and completeness.
- In the Methods section, the authors state that all enrolled patients underwent at least three hs-cTnI measurements, and the highest value was defined as the peak troponin level, regardless of the timing of the measurement. Since this is a prospective study, was there a predefined protocol or schedule for the timing of hs-cTnI tests? It would be important to clarify whether there was a standardized sampling timeline (e.g., every 1, 2, or 3 hours), as this significantly influences the identification of secondary peaks.
- The objectives of the study are not clearly defined in the Methods section. It would be beneficial for the authors to explicitly state the primary and secondary objectives, along with the outcomes evaluated (e.g., PCI complications, in-hospital mortality) and how these were prospectively measured or adjudicated.
- Please verify the percentage of excluded patients in Figure 1.
- If available, please provide information on baseline or admission medical therapy (e.g., use of antiplatelets, statins, beta-blockers, ACE inhibitors) in Table 1.
- A key piece of information missing from the manuscript is the median or mean time between AMI diagnosis (both STEMI and NSTEMI) and PCI. This time interval is crucial, especially when interpreting the troponin kinetics and the occurrence of second or third hs-cTnI peaks. The kinetic patterns may be significantly influenced by PCI timing, particularly in relation to periprocedural myocardial injury/infarction.
- In Figures 2 and 3, it is not clear what "time zero" represents. Does it refer to symptom onset, hospital admission, ICCU admission, or PCI time? This clarification is important to interpret troponin kinetics accurately and to align it with clinical events.
- The discussion should address the potential relationship between a second hs-cTnI peak and the development of periprocedural myocardial injury or infarction, which is common in AMI patients undergoing PCI. This phenomenon has been recently associated with poor prognosis (cite PMID: 39968630). Inclusion of this point would strengthen the mechanistic plausibility of your findings and highlight their clinical relevance.
- The definition of a second peak as a ≥15% increase following an initial decline is appropriate but should be more clearly justified with reference to prior studies.
- Please review the reference list for repeated citations. For example, references 10 and 20 seem to refer to the same study by van Doorn et al. (2020). Consolidating duplicates will improve the manuscript's accuracy and readability.
Author Response
"Please see the attachment."

Reviewer 2 Report
Comments and Suggestions for Authors
The study investigates hs-cTnI kinetics in ACS patients but does not present a clearly novel hypothesis or mechanistic insight. The concept of a “second troponin peak” is not well-defined or supported by prior literature. Importantly, the authors suggest an association between multiple peaks and adverse outcomes without addressing causality. For instance, in cases of reinfarction, a second troponin peak is expected as a diagnostic consequence, not a predictor. This ambiguity between marker and mechanism weakens the scientific rationale and clinical relevance.
Key definitions (troponin threshold, biphasic curves) appear arbitrary and are not supported by guidelines or references. Although regression analysis is mentioned, results are not presented in any table or figure, limiting transparency. Moreover, no ROC curves, AUC, C-index or predictive performance metrics are included. The figures provided are weak, inadequately labeled, and do not meaningfully support the hypotheses.
Despite a large cohort, the findings do not translate into actionable clinical insight. There is no demonstration of how the proposed kinetic patterns would inform decision-making, risk stratification, or therapeutic pathways.
Author Response
"Please see the attachment."

Round 2
Reviewer 1 Report
Comments and Suggestions for Authors
Thank you to the authors for the revisions made, which I believe have enhanced the quality of the final manuscript. I have no further comments.
Author Response
comment 1: Thank you to the authors for the revisions made, which I believe have enhanced the quality of the final manuscript. I have no further comments.
Replay 1: We thank the reviewers and editors for their valuable comments
Reviewer 2 Report
Comments and Suggestions for Authors
The authors have addressed several of the concerns raised in the initial review, including clarification of methodology, the addition of regression and ROC analyses, and discussion of clinical implications. While these additions improve the manuscript, a number of important issues still remain that must be addressed before the study can be considered for publication.
The manuscript continues to rely on the association between multiple troponin peaks and adverse outcomes. While the authors now reference reinfarction thresholds and clarify their approach, the manuscript still lacks a clear causal interpretation. The authors should further clarify whether they interpret second peaks as a marker, mechanism, or merely a consequence of underlying clinical processes. The distinction remains important to avoid overinterpretation.
While the authors mention that logistic regression models were performed and report AUC values for STEMI and NSTEMI subgroups, the detailed results of these models are not presented. Please include a full table listing all variables included in the models, their odds ratios, confidence intervals, and p-values to allow for proper evaluation of the findings.
Figures 2 and 3 are central to the manuscript but currently lack visual clarity and precision. The troponin kinetics curves appear overly simplistic, and labeling is insufficient. Please improve the design, resolution, and annotation of these figures to enhance interpretability and visual quality.
Although the authors have revised the discussion to include some potential clinical implications, the overall impact remains modest. The study’s findings are still primarily descriptive and exploratory. This limitation should be explicitly acknowledged in both the Discussion and Limitations sections.
